# Referral challenges and outcomes of neonates received at Muhimbili National Hospital, Dar es Salaam, Tanzania

**Mpokigwa Kiputa**[1]☯*, **Nahya Salim**[1]☯*, **Peter P. Kunambi**[2], **Augustine Massawe**[1]†

**1** Department of Paediatrics and Child Health, Muhimbili University of Health and Allied Sciences, Dar es Salaam, Tanzania, **2** Department of Clinical Pharmacology, Muhimbili University of Health and Allied Sciences, Dar es Salaam, Tanzania

☯ These authors contributed equally to this work.
† Deceased.
* mpokigwakiputa@gmail.com (MK); nsalim@ihi.or.tz (NS)

## Abstract

### Background

Functional referral system including pre referral care, access to emergency transport and ensuring continuity of care between facilities is critical for improved newborn health outcome. The neonatal transport system is quite undervalued in many sub Saharan countries, Tanzania included. This study assessed the pre referral care, transport process, ambulance characteristics, admission clinical status and outcomes of referred neonates at Muhimbili National Hospital Upanga, a tertiary facility in Dar es Salaam, Tanzania.

### Methods

A descriptive cross sectional study with a longitudinal follow up was conducted from September 2020 to February 2021 including neonates referred to Muhimbili National Hospital. A structured questionnaire was used to collect demographic characteristics and transport factors including pre referral care extracted from the referral documents and through interviewing caregivers or escorting person/nurse. Ambulances were directly observed using a structured checklist on presence, absence and functionality of supportive equipment. All enrolled neonates had a clinical assessment at admission and 48 hours post admission to determine admission clinical status and 48 hours' clinical outcome as either survived/died.

### Results

Out of the 348 neonates assessed during the study period, the median gestation age was 38 weeks (IQR 32, 39) with the mean birth weight of 2455 ± 938 g. Pre referral documentation showed that temperature was measured in 176 (57.1%), oxygen saturation and random blood glucose in only 143 (46.6%) and 116 (36.2%) neonates respectively. Ambulance was used as a means of transportation in 308 (88.5%) neonates. While no ambulance had an incubator only 7 (2.0%) neonates were kept on a Kangaroo Mother Care position. Monitoring enroute was done to only 94 (27%) of the transferred neonates with 169 (54.9%) of

**Data Availability Statement:** All relevant data are within the paper and its supporting information files.

**Funding:** This study received no financial support, it was part of the academic completion requirement supported by the Government of Tanzania through the Ministry of Health, Community Development, Gender, Elderly and Children at Muhimbili University of Health and Allied Sciences, MUHAS.

**Competing interests:** The authors have declared that no competing interests exist.

health care professionals escorting the neonates lacking training on essential newborn care. On arrival, 115 (33%) were hypothermic, 74 (21.3%) hypoxic, 30 (8.6%) with poor perfusion and 49 (14.1%) hypoglycemic. Hypothermic neonates had an increased chance of dying compared to those who were normothermic (OR = 2.09, 95% CI (1.05–4.20), p = 0.037). The chance of dying among those presenting with hypoxia was almost three times (OR = 2.88, 95%CI (1.44–5.74), p = 0.003) while those with poor perfusion was almost five times (OR = 4.76, 95%CI (1.80–12.58), p = 0.002). Additionally, neonates who had hyperglycemia (RBG > 8.3mmol/l) on arrival had a higher probability of dying compared to those who were euglycemic [(OR = 3.10, 95% CI (1.19–8.09) p = 0.021]. Overall mortality was 22.4% within 48 hours of admission and risk of dying increased as the presence of poor clinical status added on.

## Conclusion

Neonatal transportation in Dar es Salaam, Tanzania was observed to be challenging. Pre transfer care and monitoring during transportation was inadequate and this contributed to poor clinical status on admission. Hypothermia, hypoglycemia, hyperglycemia, hypoxia and poor perfusion on admission were associated with increased mortality. Effective referral network is needed for improved neonatal health outcomes. Pre referral supportive care, training of health care professionals, transportation with improved monitoring, clear communication protocol and referral documentation should be invested and effectively utilized.

## Introduction

Globally, neonatal mortality rate has decreased from 36.6 deaths per 1000 live birth to 18.0 deaths per 1000 live birth between the year 1990 and 2017 [1]. The rate of decline in sub Saharan Africa needs to be accelerated to achieve the Sustainable Development Goals (SDG's) by 2030 [1]. In Tanzania, the current neonatal mortality rate is estimated to be 25 deaths per 1000 live birth [2], with institutional deliveries of 76% according to District Health Information Software (DHIS2) of 2018. Considerable efforts have been emphasized on quality facility based care including investment in network of care to facilitate survival [3]. A large number of inter facility transfer is already happening and it is likely to increase.

Referral system challenges such as inadequate transportation, lack of communication, poor documentation and lack of monitoring have been established as factors impeding the stride towards reducing neonatal mortality in developing countries [4]. A well-established referral system is key to transfer neonates to a tertiary care facility [5–7].

Transportation of a neonate from one facility to another under ideal conditions is still a challenge in many developing countries, Tanzania included. Most of the neonates arrive in poor clinical conditions, which are mostly preventable. Previous studies have shown that hypothermia, hypoglycemia, poor perfusion and hypoxia are associated with increased mortality among transported neonates [8–12].

There is enough evidence to support that transport by a skilled organized team reduces neonatal morbidity and mortality [9, 13, 14]. The goal of all neonatal transport teams should be transporting a well-stabilized neonate. Pre transport stabilization is crucial; this entails securing the patency of the airway, breathing and circulation. Pre-transport procedures such as establishing an intravenous access should be carried out before arrival of the transport team [15, 16] and continuous monitoring is needed on the way to a higher facility.

Despite the effort to strengthen the referral system in the country and the improvement of health care delivery in terms of infrastructure and manpower, the outcome of referred neonates is still poor. According to raw data collected in 2020 from the tertiary facility, Muhimbili National Hospital (MNH), neonatal mortality among the out-borns is almost twice as high compared to the in-borns admitted at the same neonatal unit.

This study aimed at describing the transport characteristics (i.e. mode of transport, equipment, communication, accompanying personnel, pre referral care and monitoring enroute) for the referred neonates. Additionally, it sought to assess the clinical status at admission in terms of the presence of hypothermia, hypoxia, perfusion and hypoglycemia for the transported neonates to Muhimbili National Hospital (MNH). Furthermore, it determined 48 hours' outcome and its associated factors in relation to the admission clinical status of referred neonates.

### Ethics statement

Permission to carry out the study was sought from Muhimbili University of Health and Allied Sciences (MUHAS) Institutional Review Board (IRB) with an approval number MUHAS-REC-07-2020-306. Written informed consent to participate in the study was sought from both the parent/guardian and health care personnel who escorted the neonate prior to any study procedure. Illiterate parents/ guardians were asked for a witness who participated within the discussion prior to obtaining their thumbprints and witness signature. All neonates received appropriate treatment according to the national treatment guidelines of Tanzania regardless of participation.

## Materials and methods

### Study area

Muhimbili National Hospital (MNH), Upanga is a tertiary referral centre and teaching hospital receiving patients from the five municipalities in Dar es Salaam (Ilala, Kinondoni, Kigamboni, Ubungo and Temeke) and other upcountry regions. It has a transitional level 2 plus Neonatal Intensive Care Unit (NICU) for caring critically ill neonates. Staff cadres constitutes of 2 neonatologists, 9 pediatricians, 5–10 residents at a time, 3 medical registrars, 9–11 interns and 59 trained nursing staff. The referral annual admissions range between 1440 and 3360 neonates.

### Study design

A descriptive cross sectional study with a longitudinal follow up was carried out at Muhimbili National Hospital (MNH), in Dar-es-Salaam region, Tanzania.

### Participant recruitment and data collection

All neonates referred to MNH from September 2020 to February 2021 were eligible for enrollment except for neonates with obvious congenital anomalies. A consecutive sampling was used to recruit referred neonates until the required sample size of 349 neonates was met. Sample size was calculated using the expected proportion of referred neonates transported to the neonatal unit of MNH who died in the first week post admission. This was established using a pilot study conducted at MNH prior to this, and was found to be 35%. A sample size of 349 was achieved using the Kish Leslie formula with 95% confidence level and 5% margin of error.

Data was collected day and night using a pretested structured questionnaire designed for the study (S1 File). Data on demographic characteristics of the study participants and pre referral treatment if any were obtained from the referral documents and by interviewing the caregiver or escorting personnel/nurse. The escorting health care personnel was enquired

about the transport process including pre transport care and monitoring during transport. Escort team were also asked on whether they received a training on essential newborn care. Data on ambulance characteristics was obtained through direct observation of the transport by looking inside the transport for the presence of necessary equipment using WHO neonatal ambulance checklist [17]. Gestation age, birth weight and date of birth were extracted from the referral documents. Clinical status on admission was measured using **TOPS** model [18–20], whereby;

**T**emperature was recorded in degrees centigrade and hypothermia defined as an axillary temperature on admission < 36.5˚C measured with a digital thermometer (KONIG HC-DT10 Digital Thermometer, United Kingdom). Fever as axillary temperature on admission > 37.5˚C.

**O**xygen saturation (SPO2) was measured by using the pulse oximeter (ChoiceMMEd fingertip pulse oximeter, manufacturer: Beijing choice electronic technology Co., Ltd. Fuxing road A36 Beijing, 100039 China). Hypoxia was defined as oxygen saturation less than 90% in room air [21].

**P**erfusion, Capillary Refill Time was recorded by applying gentle pressure on the sternum for 3–5 seconds to cause blanching then the time taken for color to return was measured.

**S**ugar, Random Blood Glucose was measured by using a haemoglucometer machine (STANDARD™ GlucoNavii® GDH. Manufacturer: SD Biosensor, Inc. Gyeonggi-do, 16690, Republic of Korea). Hypoglycemia was defined as the blood glucose measurement of less than 2.5 mmol/L and hyperglycemia as blood glucose measurement greater than 8.3 mmol/L.

Neonates who were found to be hypoglycemic on admission were given bolus intravenous dextrose, those who were hypoxic received supplemental oxygen according to their needs and those who were hypothermic were kept in a warmer. Neonates who were in shock upon arrival were resuscitated with intravenous fluids according to the hospital protocol. There was room for escalation of care for neonates who needed advanced respiratory support such as ventilating machine.

All enrolled neonates were assessed at 48 hours post admission to capture the early outcome that was defined as survived or died.

## Data management and statistical analysis

Data analysis was done using SPSS version 23.0. Dependent variables were admission clinical status as per TOPS model and outcome defined as survival or death of referred neonates at 48 hours post admission. Independent variables were 1. transport factors including mode of transport, presence/functionality of equipment in the transport, accompanying personnel cadre, training status of escorting personnel, monitoring during transport, and pre transport care provided and prior communication 2. Demographic factors: age, sex, birth weight, gestational age, age in days, mode of delivery, Apgar score.

Continuous variables for social-demographic and clinical characteristics of study participants were described using mean and standard deviations or medians and interquartile range depending on normality of distribution. Proportions were presented in bar graphs. Factors with P < 0.2 from univariate analysis were included in the multivariable analysis. Multiple logistic regressions (odds ratio) was used to determine association with clinical outcome. A p-value ≤ 0.05 was considered statistically significant at 95% confidence interval.

## Results

### Baseline characteristics of the study participants

A total of 349 referred neonates were recruited during the study period, one was excluded from the study due to a congenital anomaly which was incompatible with life

(anencephaly). Fig 1 shows the proportion of participants received at Muhimbili National Hospital (MNH), Upanga stratified by referring districts of Dar es Salaam, Tanzania. Out of the 348 neonates, 207 (59.5%) were term babies while 141 (40.5%) were preterm. Majority were male babies 197 (56.6%) with 258 (74.1%) neonates delivered by spontaneous vaginal delivery (SVD) while 85 (24.4%) were born via caesarean section. The median gestation age was 38 weeks (IQR; 32–39) with the mean birth weight of 2455 ± 938 g. The mean age at admission was 1.78 ± 0.757 days. Table 1 describes the socio- demographic and clinical characteristics of the study participants.

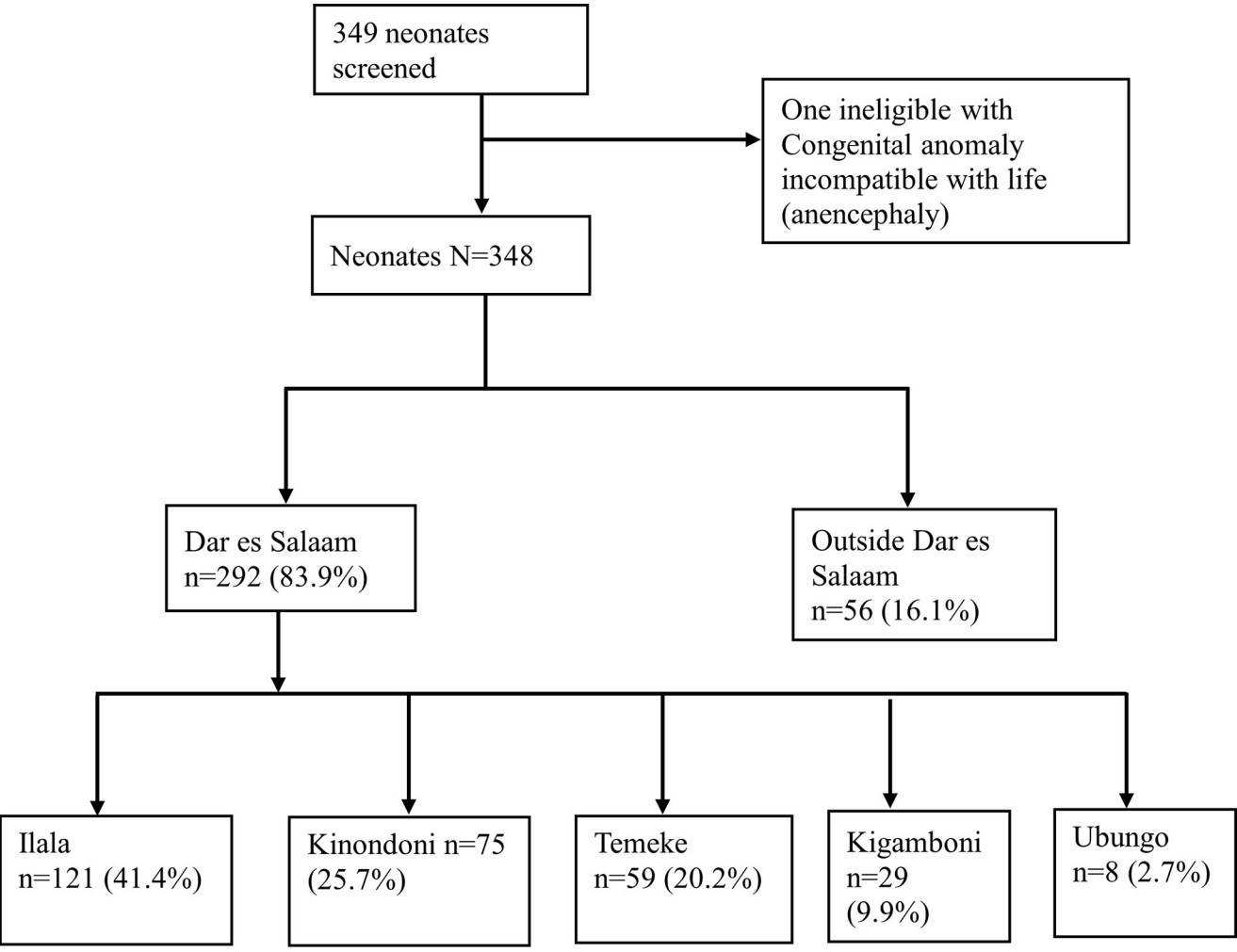

**The % contribution were as follows:**
Public hospital 89.4%
Private hospital 9.2%
Home 1.4%

**Fig 1. Flow of the study participants referred from various districts of Dar es Salaam, Tanzania.**

**Table 1. Socio-demographic and clinical characteristics of study participants, N = 348.**

| Characteristic | Category | Frequency (n) | Percent (%) |
|---|---|---|---|
| **Sex** | Male | 197 | 56.6 |
| | Female | 151 | 43.4 |
| **Age at admission (days)** | | | |
| | 0–1 | 229 | 65.8 |
| | 2–6 | 95 | 27.3 |
| | 7–28 | 24 | 6.9 |
| Mean age at admission (SD) (days) | | 1.78 (± 0.757) | |
| **Gestational age (weeks)** | <28 | 25 | 7.3 |
| | 28–32 | 68 | 19.7 |
| | 33–36 | 45 | 13.0 |
| | >37 | 207 | 60.0 |
| Median gestational age (week) (IQR) | | 38 (32–39) | |
| **Birth weight (g)** | Less than 1000 | 21 | 6.0 |
| | 1000–1499 | 37 | 10.6 |
| | 1500–2499 | 96 | 27.6 |
| | 2500+ | 194 | 55.8 |
| Mean birth weight (SD) (g) | | 2455 (± 938) | |
| **Mode of delivery** | SVD | 258 | 74.1 |
| | Breech delivery | 4 | 1.2 |
| | Vacuum delivery | 1 | 0.3 |
| | Caesarean section | 85 | 24.4 |
| **Apgar score at 1st and 5th minutes** | 0–3 (low) | 4 | 1.1 |
| | 4–6 (moderately abnormal) | 65 | 18.8 |
| | 7–10 (reassuring) | 279 | 80.1 |

## Pre transport care and documentation at referring facility

According to data extracted from the referral forms, two hundred and forty-six (79.9%) neonates had an intravenous line inserted prior to transfer. Temperature was measured in 176 (57.1%) neonates before transfer. Oxygen saturation and random blood glucose were measured pre transfer in only 143 (46.6%) and 116 (36.2%) neonates respectively.

## Transport process

**Communication.** Out of 348 neonates who were transferred to the neonatal unit of MNH, in only 26.7% the notification was given prior to transfer.

**Escorting personnel.** Out of the 340 neonates who were escorted, two hundred and thirty neonates (66.1%) were escorted by the registered nurse. Seventy-five (21.6%) were escorted by the nurse attendant while eight (2.3%) were escorted by a doctor and thirty-two (9.2%) were escorted by the family member. Of the health care professional escorting the neonate, two hundred and nine (66.1%) had no training on essential newborn care.

**Access to ambulance service.** Majority of the neonates were transported by ambulance 308 (88.5%), followed by private car/taxi 30 (8.6%), public service vehicle 9 (2.6%) and tricyclic motor vehicle 1 (0.3%).

**Monitoring during transport.** Monitoring was done to only 94 (27%) of the transferred neonates. Of those 69 (73.4%) had their temperature measured. Capillary refill time was monitored in 37 (39.4%) of the transferred neonates while random blood glucose was checked in 27 (28.7%). Just 57 out of 94 (60.6%) had their oxygen monitored during transport.

**Warmth during transport.** Most of the referred neonates were kept warm using local clothes (Khanga/Kitenge), 233 (67.0%). Only 7 (2.0%) were kept on a Kangaroo Mother Care position, while no neonate was kept in an incubator during transfer.

**Reasons for referral.** The main reason for transfer was to seek specialized care 164 (47.1%) followed by a lack of newborn unit 65 (18.7%), referred for further investigation 65 (18.7%), lack of equipment 23 (6.6%) and lack of personnel 7 (2.0%).

## Ambulance characteristics

For those who came with an ambulance 308, 294 (95.5%) had an oxygen supply, but in 42.5% of those the oxygen delivery system was not functioning. Only 201 (65.3%) had resuscitation equipment such as Ambu bag packed in the ambulances. Resuscitation drugs such as adrenaline were included in 200 (64.9%). Monitoring equipment were not available in 104 (33.8%) ambulances. IV fluids were present in 237 (77.0) while only 184 (59.7%) had suction apparatus. No ambulance had an incubator.

Fig 2 summarizes coverage gap during the referral journey of a newborn in Dar es Salaam, Tanzania.

## Admission clinical status and outcome at the receiving facility, MNH

Seventy-eight neonates (22.4%) died within 48 hours of admission. On arrival, the clinical status of the neonates were as follows; 115 (33%) hypothermic, 74 (21.3%) hypoxic, 30 (8.6%) with poor perfusion and 49 (14.1%) hypoglycemic. Those with hypothermia ($< 36.5°C$), were two times more likely to die compared to those who were normothermic (OR = 2.09, 95% CI (1.05–4.20), p = 0.037). Thirty-eight (50.7%) of those who had hypoxia (SPO2 $<$ 90%) at arrival died compared to thirty-seven (49.3%) who survived (OR = 2.88, 95%CI (1.44–5.74), p = 0.003). Twenty-one (67.7%) neonates with prolonged capillary refill time (CRT) died compared to ten (32.3%) who survived (OR = 4.76, 95%CI (1.80–12.58), p = 0.002). Out of forty-nine neonates who arrived with hypoglycemia (RBG $<$ 2.5mmol/l) twenty-one (42.9%) died in the first 48 hours (OR = 2.13, 95%CI (0.96–4.74), p = 0.064). Additionally, neonates who had hyperglycemia (RBG $>$ 8.3mmol/l) on arrival were three times more likely to die compared to those who were euglycemic [(OR = 3.10, 95% CI (1.19–8.09) p = 0.021]. Table 2 summarizes the results of univariate and multivariable analysis of factors associated with mortality.

The combined effect of poor clinical status on admission in terms of hypoglycemia, hyperglycemia, hypoxia, prolonged CRT and hypothermia were associated with increased mortality as shown in Fig 3.

## Discussion

Despite its importance, neonatal transportation has often been overlooked in resource constrained and poor countries like Tanzania [16, 22, 23]. Due to lack of organized neonatal transportation system, most of these neonates arrive in poor clinical condition which influence their outcome negatively [5, 8]. This study assessed the gaps on pre referral care, neonatal transportation, admission clinical status and outcome of the referred neonates at MNH–Upanga, a tertiary facility in Dar es Salaam, Tanzania.

In our study, nineteen percent of the neonates were referred due to lack of a newborn unit in their facilities, seven percent had no equipment while almost half of others were referred for specialized medical or surgical care. Improving level of care in institutions including establishment of well-equipped neonatal units could reduce referrals and mortality of newborns. The ministry of health, Tanzania in collaboration with the Newborn Essential Solutions and Technologies (NEST360) program have been improving neonatal infrastructures, installing

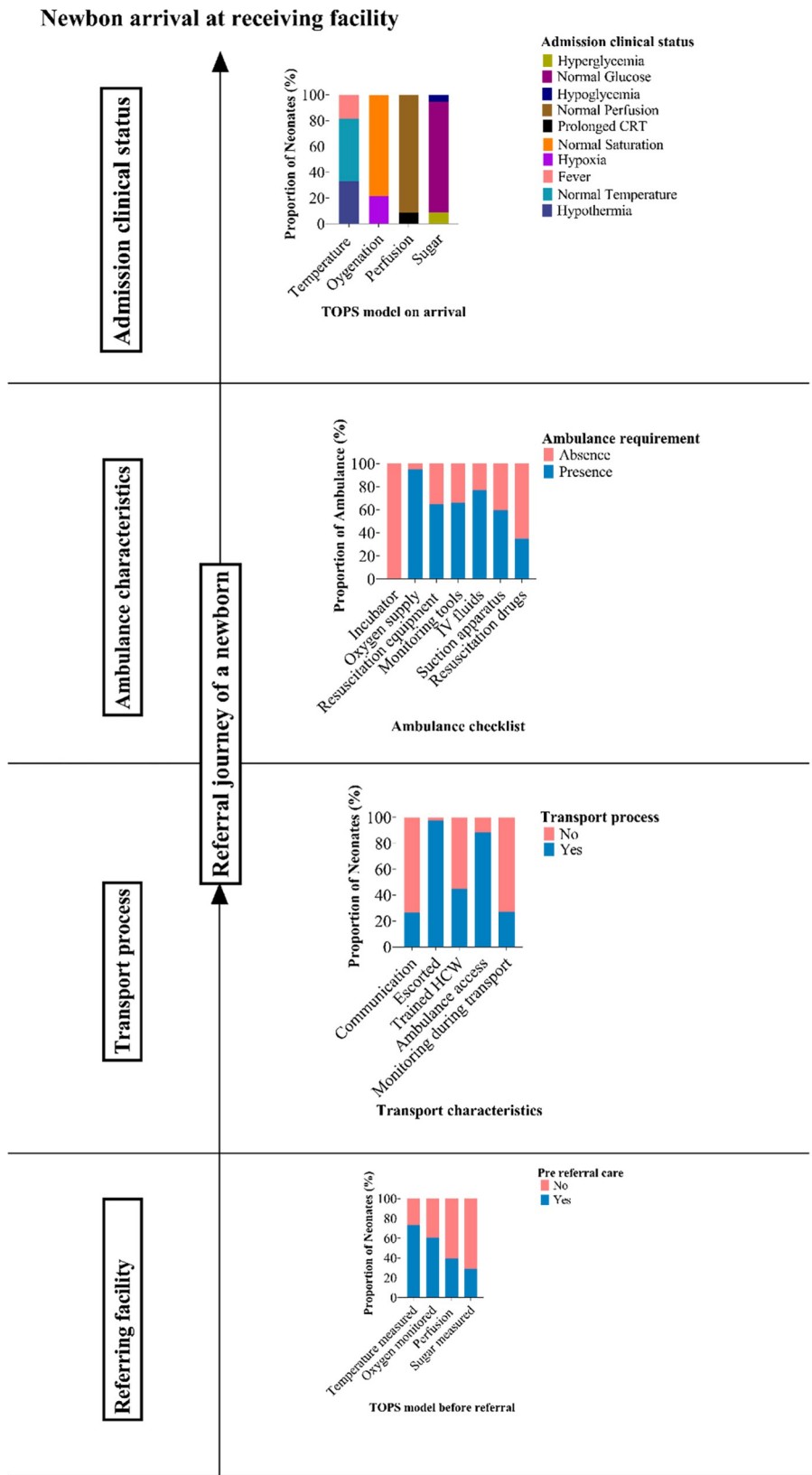

**Fig 2. Coverage gap during the referral journey of a newborn in Dar es Salaam, Tanzania.**

**Table 2.** Univariate and Multivariate analysis of factors associated with mortality.

| Variable | Univariate analysis | | | Multivariable analysis | | |
|---|---|---|---|---|---|---|
| | cOR | 95% CI | P -value | aOR | 95% CI | P -value |
| **Gestational age (weeks)** | | | | | | |
| < 28 | 7.95 | 3.19–19.83 | **< 0.001** | 3.16 | 0.77–12.99 | 0.111 |
| 28–33 | 1.10 | 0.57–2.15 | 0.769 | 0.61 | 0.22–1.74 | 0.357 |
| 34–36 | 0.49 | 0.18–1.30 | 0.151 | 0.35 | 0.11–1.17 | 0.089 |
| ≥ 37 | Ref | | | | | |
| **Weight (Kg)** | | | | | | |
| < 1.0 | 4.65 | 1.86–11.59 | **< 0.001** | 1.99 | 0.45–8.74 | 0.362 |
| 1.0–1.4 | 3.03 | 1.43–6.41 | **0.004** | 2.54 | 0.70–9.24 | 0.158 |
| 1.5–2.4 | 1.07 | 0.57–2.02 | 0.829 | 1.56 | 0.68–3.60 | 0.293 |
| ≥ 2.5 | Ref | | | | | |
| **Temperature ($^0$C)** | | | | | | |
| Hypothermia (< 36.5) | 3.65 | 2.04–6.53 | **< 0.001** | 2.09 | 1.05–4.20 | **0.037** |
| Fever (> 37.5) | 1.62 | 0.76–3.43 | 0.209 | 1.25 | 0.53–2.96 | 0.612 |
| Normal (36.5–37.5) | Ref | | | | | |
| **Capillary Refill Time** | | | | | | |
| Prolonged (> 3 seconds) | 9.58 | 4.28–21.44 | **< 0.001** | 4.76 | 1.80–12.58 | **0.002** |
| Normal (< 3 second) | Ref | | | | | |
| **Oxygen saturation** | | | | | | |
| Hypoxia (< 90%) | 5.98 | 3.41–10.51 | **< 0.001** | 2.88 | 1.44–5.74 | **0.003** |
| Normal (≥ 90%) | Ref | | | | | |
| **Random blood glucose (mmol/L)** | | | | | | |
| Hypoglycemia (< 2.5) | 4.19 | 2.17–8.07 | **< 0.001** | 2.13 | 0.96–4.74 | 0.064 |
| Hyperglycemia (> 8.3) | 6.87 | 3.08–15.36 | **< 0.001** | 3.10 | 1.19–8.09 | **0.021** |
| Normal (2.5–8.3) | Ref | | | | | |

Key. cOR: crude odds ratio, aOR: adjusted odds ratio, Ref: Reference category.

equipment and providing in service training to strengthen neonatal care in hospitals across three regions (Dar es Salaam, Mbeya and Moshi) since early 2020. The phase I implementation of NEST 360 program is focusing on regional and tertiary facilities in a stepped wedged approach. While it's clear that the program is influencing the neonatal care, it's too early and challenging to link the measurements due to lack of baseline data. Currently, all the three regional referral hospitals in Dar es Salaam have been upgraded to level 2 plus with ongoing quality improvement and mentorship within the region. The findings of this study may provide highlights on baseline data for future studies. Strengthening harmonized health services and investing in coordinated network of care is of paramount importance to ensure safe transfer and survival for those in need.

Pre transport care is pivotal before transfer to a higher facility and is generally agreeable that for a quality transport and desirable outcome, stabilization prior to transfer is mandatory [24]. A study by Narang et al. demonstrated an overall decrease in mortality among neonates in whom lifesaving interventions were done pre referral [25]. In our study, we focused on TOPS model and we noted that temperature was measured in more than half of the neonates prior to transfer. In more than half of the referred neonates neither oxygen saturation nor random blood glucose was measured or documented. Twenty percent of the neonates were transferred without a secured intravenous line which is vital for institution of drugs during resuscitation. In the majority, there was no documentation of what was done, for example to

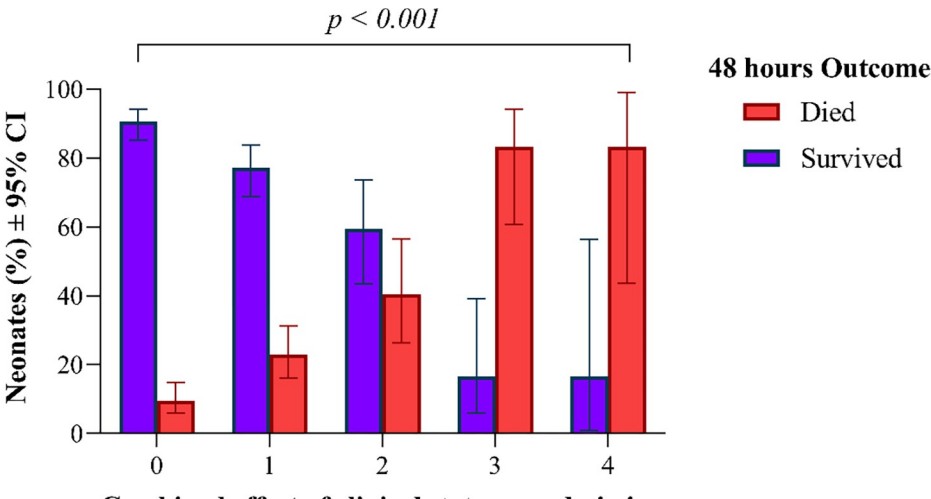

0 – Neonates without any poor clinical status on admission
1 – Neonates with one of the poor clinical status on admission
2 – Neonates with two of the poor clinical status on admission combined
3 – Neonates with three of the poor clinical status on admission combined
4 – Neonates with all four poor clinical status on admission combined

NOTE: Clinical status defined as presence or absence of Hypothermia, Hypoxia, Poor Perfusion, Hypoglycemia

**Fig 3. Combined effect of exposures on newborns mortality outcome at 48 hours post admission at Muhimbili National Hospital.**

those who were found to be hypothermic. It is clear that poor pre transport care contributed to poor conditions at arrival with subsequent poor outcome. These findings have been documented in previous studies [26, 27]. Knowledge, skills and competence of health professionals is key to saving lives at first level referral facilities, this gap need further attention toward quality newborn care.

Three quarters of the referred neonates were brought in without any notification given to the neonatal unit that could give room for preparation and advice. This finding is similar to the study done by Abdulraheem et al. in Ibadan Nigeria which showed no communication or documented information regarding the clinical stability of any of the neonates prior to transfer [16]. In Tanzania, the referring institution is tasked with providing the escorting personnel and pre referral communication of the case. Our study has shown that in addition to lack of communication, most of the times staff who will render minimal interruption to the hospital running are utilized rather than competent nurses who can support a neonate in the ambulance. Quite a number of neonates were escorted by nurse attendants 73 (21.5%) who were not trained to provide that level of care. Furthermore, this study found more than half of the escorting health care personnel lacking training on essential newborn care. A study done in Jamaica reported one in three of the personnel accompanying the neonate were not skilled in neonatal resuscitation [15]. Transfer of neonates by trained personnel has shown to reduce morbidity and mortality [28]. Improvement in communication system and skills on emergency care is needed.

The preponderance of neonates who were referred to our facility came with hospital organized ambulances at 308 (88.5%) compared to other modes of transport. This is different from studies done in Ghana and Nigeria whereby majority of neonates were brought in by taxis and private vehicles at 36% and 43.9% respectively [16, 23]. The findings are consistent with the

study done in the EThekwini health district of KwaZulu-Natal, South Africa whereby all the referrals were transported by ambulance [29]. In Tanzania, ambulance services are cost free in all government owned facilities this could explain why many neonates were brought in using ambulances. Nonetheless, it is worth noting that despite the availability of ambulance, inadequate equipment and functionality were observed. Lack of equipment such as patient monitors impedes actions that could be taken to reverse deterioration during transport. It is not surprising that many neonates arrived at the tertiary hospital in unsteady clinical conditions i.e poor perfusion, hypoxic, hypoglycemic and hypothermic.

Most ambulances had oxygen supply similar to a study done in Bangladesh [30], in our study only 57.5% of the oxygen delivery system were functional. This fact plus lack of monitoring during transport clearly could have contributed to a number of neonates arriving hypoxic at the facility. Similar to our findings, a study by Mehta et al. done in India using the TOPS model established that neonates who had hypoglycemia and prolonged capillary refill time on admission had an overall poor outcome compared to their counterparts [19]. Additionally, we found a number of neonates who were hyperglycemic, had significant higher odds of mortality. These could have been neonates whose blood sugar levels were not monitored and some were given continuous infusion of glucose without any blood glucose check. Monitoring during transportation is critical, ambulance services need to be coordinated.

Dar es Salaam has a tropical climate located 16m above sea level with an average annual temperature of 26.1˚C. Despite this fact, one third of the neonates were hypothermic on admission which was associated with increased mortality. This finding is similar to studies done in low resource settings in Ghana and India where almost half of the neonates were found to be hypothermic [23, 25]. A study which was done on the same setting in 2003 by Manji et al. found a threefold increase in mortality and morbidity among hypothermic neonates [11]. Hypothermia in our study was contributed by lack of monitoring during transport and inferior means of maintaining warmth as more than two third of the referred neonates were covered by only light local clothes such as khanga and kitenge. Most facilities could not afford incubators in their ambulance, nevertheless the practice of KMC which has been shown to be cheap and effective in preventing hypothermia was observed in only 2% of the transferred neonates. In a study conducted by Rathod et al., in southern India, KMC practice was not observed at all [22].

The overall mortality of neonates 48 hours post admission was 22%. Furthermore, a correlation between poor clinical status on admission in terms of hypoxia, hypothermia, hyperglycemia and poor perfusion with increased risk of mortality 48 hours post admission has been demostarted in previous studies [9–12]. For example, persistent hypothermia in newborns can lead to complications such as infection, hypoglycemia, and metabolic acidosis, and increase the risk of late-onset sepsis [31, 32]. Managing hypothermia and maintaining a healthy temperature in neonates is essential to survival of the particularly vulnerable newborn [31, 33]. All these are preventable conditions, if tackled appropriately could increase survival of the newborns.

Overall, this study demonstrates inadequate pre transport care, documentation and lack of monitoring enroute contributing to poor clinical conditions of neonates at admission. Emphasis has to be placed on having coordinated ambulance services, strengthening referral system and pre referral care to neonatal transportation. Additionally, ensuring continuous training to health care workers on neonatal resuscitation and essential newborn care is key. Regional and national coordination is needed to strengthen the referral system network.

Our study may have been exposed to recall bias as some of the information relied on the escorting health personnel. Hawthorne effect can't be ignored as the behaviors might have changed on the course of the study including improvement in the referral and transportation

of neonates as they become aware they are being observed. Limitation in recording full data due to poor/incomplete documentation based on the referral forms, consequently contributed to the inadequate data of pre-referral care assessment, reasons for referral and monitoring enroute information. Despite the limitations, the findings reflect the gaps of referral system which need to be improved for quality care of newborns at the primary and secondary levels. Further studies are required to learn on barriers and facilitators of effective referral at all levels of care. All efforts should be done to ensure every small and sick newborn with condition(s) that cannot be managed effectively with available resources receive appropriate, timely referral through integrated newborn service pathways with continuity of quality care, including provisional of warmth using cost effective measures such as KMC during transportation.

## Supporting information

**S1 File. Structured questionnaire English version.**
(DOCX)

**S1 Data.**
(XLSX)

## Acknowledgments

The authors acknowledge all staff at Muhimbili National Hospital, all caregivers/guardians of neonates, escorting persons and nurses who made this study possible to accomplish what we have. We are particularly obliged to Dr. Robert Moshiro and all members of Peadiatrics and Child Health Department at MNH and MUHAS. We would like to dedicate this work to our mentor, neonatologist, a father and a neonatal champion in Tanzania, the Late Dr. Augustine Massawe. May his soul rest in peace, Amin.

## Author Contributions

**Conceptualization:** Mpokigwa Kiputa, Augustine Massawe.

**Data curation:** Mpokigwa Kiputa, Peter P. Kunambi.

**Formal analysis:** Peter P. Kunambi.

**Investigation:** Mpokigwa Kiputa.

**Methodology:** Mpokigwa Kiputa.

**Software:** Peter P. Kunambi.

**Supervision:** Nahya Salim, Augustine Massawe.

**Writing – original draft:** Mpokigwa Kiputa, Nahya Salim.

**Writing – review & editing:** Mpokigwa Kiputa, Nahya Salim, Augustine Massawe.

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
