## [Decision Letter · Decision Letter 0]

2 Feb 2022

PONE-D-21-40535Referral Challenges and Outcomes of Neonates Received at Muhimbili National Hospital, Dar es Salaam, Tanzania.PLOS ONE

Dear Dr. KIPUTA,

Thank you for submitting your manuscript to PLOS ONE. After careful consideration, we feel that it has merit but does not fully meet PLOS ONE’s publication criteria as it currently stands. Therefore, we invite you to submit a revised version of the manuscript that addresses the points raised during the review process. Besides the important reviewers' comments, authors have to address the following points:Authors are encouraged to provide unidentified patients data. According to PLOS ONE publication criteria, authors are required to make all data underlying the findings described fully available, without restriction, and from the time of publication. PLOS allows rare exceptions to address legal and ethical concerns, that should be clearly explained.Authors reported only selected factors associated with mortality. However, authors should analyze/discuss several other potential factors, such as gestational age, gender, mode of delivery, birth weight, transport process, mode of transportation, reasons for referral, maternal age, Apgar score, travel time, and presence of antenatal complications.Authors should develop proper and specific conclusions (based on study findings).

We look forward to receiving your revised manuscript.

Kind regards,

Elsayed Abdelkreem, MD, PhD

Academic Editor

PLOS ONE

Journal Requirements:

Reviewers' comments:

Reviewer's Responses to Questions

**Comments to the Author**

1. Is the manuscript technically sound, and do the data support the conclusions?

Reviewer #1: Yes

Reviewer #2: Partly

Reviewer #3: Yes

Reviewer #4: Partly

2. Has the statistical analysis been performed appropriately and rigorously? 

Reviewer #1: Yes

Reviewer #2: Yes

Reviewer #3: Yes

Reviewer #4: No

3. Have the authors made all data underlying the findings in their manuscript fully available?

Reviewer #1: No

Reviewer #2: Yes

Reviewer #3: Yes

Reviewer #4: Yes

4. Is the manuscript presented in an intelligible fashion and written in standard English?

Reviewer #1: Yes

Reviewer #2: No

Reviewer #3: Yes

Reviewer #4: No

5. Review Comments to the Author

Reviewer #1: Referral Challenges and Outcomes of Neonates Received at Muhimbili National Hospital, Dar es Salaam, Tanzania

PONE-D-21-40535

I would like to congratulate the authors for a good work done. I have some minor concerns which may improve upon the quality of the manuscript.

1. Introduction: lines 72-73 stated “About half of neonates die within the first 24 hours after birth and 75% in the first seven days of life” Please check this information and present it appropriately. You mean; about half of all neonatal deaths occur within the first 24 hours……

2. Participant recruitment and data collection: line 126 stated that: “Data was collected day and night…”. Were there any difference in outcome when day or night referrals were considered

3. Neonatal age in days at time of admission was not reported. This is important to understand the outcome measure presented in the study

4. Results section monitoring during transport: lines 183-184: “Blood circulation was monitored in 37 (39.4%) of the transferred neonates…..”. How was it done and how was this assessed.

5. Table 1: presented “Apgar score 1st and 5 minutes” however only single measurement was presented. A little explanation would dispel the confusion.

Reviewer #2: General comment

This would have been an important study that contributes to the understanding of the Referral Challenges and Outcomes of Neonates Received at Muhimbili National Hospital, Dares Salaam, Tanzania. However, there are minor flaws in the method of the study, discussion and grammar that need a revision before publication.

Specific comments

1. In abstract section, it needs more elaboration in why the study is needed/ gap of the study, As much as possible avoid acronyms in the abstract and rewrite the abstract in very short and precise way particularly in the result section.

2. Pay great attention for punctuations.

3. In introduction section, would start with the global and regional/ continent evidence on neonatal mortality and relate it with how referral contributes. One paragraph which describes why this study is needed. Generally, introduction lacks coherence of ideas, world to Tanzania figure of neonatal mortality, gap of the study, clear description of clinical outcome of referral neonates in previous study. Introduction needs substantial modification.

4. In materials and methods,

Study area: Add the number of physician, nurses and other health care professionals working in neonatal ward and NICU. It gives an insight for why high burden of neonatal death; Is because of shortage of HCW or not? You stated the number of admission but we can’t get patient-to-physician proportion.

• In line 147, it states as “All enrolled neonates were assessed at 48 hours post admission to capture the early outcome that was defined as survived or died” what is your baseline reference? If there is, cite it.

5. Result

• In line 157, it states as “ A total of 349 referred neonates were screened during the study period, one was excluded from the study due to a congenital anomaly which was incompatible with life (anencephaly).” How it could be? You already excluded the known congenital anomalies before including in the study.

• In line 161, “258 (74.1%) were delivered by Spontaneous Vertex Delivery (SVD) while 85 (24.4%) were born via caesarean section.” Rewrite this.

• What is the rationale behind to use median/mean and IQR and SD? You used median for GA and mean for BWt….

• In line 176, it states as “thirty-two (9.4%) were escorted by the family member.” In method section, you stated that the pre transfer data and transportation data/monitoring during transportation were taken from accompanying personnel/escort personnel. My question is how did you collect data from those who escorted by the family member? They don’t know about the pre transfer treatments and monitoring process.

• In line 208-211, Twenty-one (67.7%) neonates with prolonged capillary refill time (CRT) died compared to ten (32.3%) who survived (OR=4.76, 95%CI (1.80-12.58), p=0.002)”. This sentence is hard to follow. Consider revision. Either describes using odds ratio or proportion in the whole document.

6. The discussion is inadequate and need to be revised thoroughly. Your independent variables were not well stated as compared to other findings with possible justification. Contributing factors are not discussed in detail as compared to previous studies. It doesn’t tell us the discovery and how these results contribute to the current debate and policies in the research specific context and beyond.

In table 2: Is you used univariate or bivariate? Which one do you think appropriate? What is difference b/n multivariate analysis and multivariable analysis? Which one is appropriate for your study?

Generally, give much concern for grammar, punctuation, capitalization and flow of ideas.

Reviewer #3: The study titled Referral Challenges and Outcomes of Neonates Received at Muhimbili National

Hospital, Dar es Salaam, Tanzania explored problems for preparation, care during transport and outcome in terms of death with in 48 hours, a crucial issue that are under addressed in countries like Tanzania. This findings will guide to plan for way forward to achieve SDG.

in line 180 , spelling mistake, seems it will be taxi. others seem good

Reviewer #4: Dear authors,

This manuscript describes the outcome of neonates referred to tertiary care hospital and identification of underlying factors for poor outcome. The topic is very important particularly from point of view of high NMR in developing countries. However, there are some serious flaws in the paper which need to rectify before further processing:

First of all, the authors need to work on the English language written in the paper for grammar, and syntax of the sentences.

Please clarify that whether all babies were born at hospitals, if it is so, then, please elaborate the infrastructure available at these peripheral hospitals where child births have taken place. Please add whether these neonates were referred from a single health facility or multiple hospitals.

Also, please add following information:

Referral indications: Please describe the approach used to determine indication for referral. Please explain whether this information was extracted from cards with details of parameters used in referral cards. Sometime, the indications listed are not necessarily mutually exclusive” e.g. need for mechanical ventilation may overlap with prematurity or birth asphyxia. If there was more than one indication, how was this treated analytically?

What definitions did you use for classification of period of gestation, birth asphyxia, and severity of hypothermia (like moderate or severe?)

Gestational age is frequently poorly recorded and/or women present late to care. The accurate determination of gestational age or classification of prematurity is critically important as your results present approximately 40% prematurity. Please identify this specifically for weight as well – it is extremely important to know if these were scale weights at birth and what was done for home births (if any).

The authors have not included any information about the comparison between the referred neonates that lived and those that died. Please add information that describes this comparison.

Do you have any information about timeliness of referrals or care provided before referrals? In final discussion and conclusions, you have mentioned that the peripheral center and pre-referral stabilization is needed as well as upgrades. While this may be true, you do not present sufficient information in the manuscript to lead to this recommendation. Also explain about the duration of stay for neonates at referring facility and treatment provided before referral.

Please add duration transport as well distance travelled by neonates and perform statistical analysis of this factor also.

As the sample size 348, in order to know if this sample is representative for the whole study population, it would be needed more information regarding hospital and setting. What is the population size attended by this hospital, how many neonatal admission by year this hospital has and how many are inborn and how many outborn. This would give us a roughly idea if the sample size is enough to be representative for the whole population. Also what is the NMR in this hospital or at least in the region (authors have given the national NMR but likely the regional is different). Please also write exact number of the neonates at every place where percentage is used.

Explain in a better way how the factors affecting the outcome of neonates were identified. What kind of analysis was done to determine significance of parameters? The significance level indicated was for what? Whether only those parameters having significant values on univariate analysis were included in multivariate analysis?

In LMICs, antenatal screening is one of the important factors in identification of high risk pregnancies, but surprisingly this information was missed in manuscript.

There are no details of referral challenges that might be faced during referral of neonates in the region as you have stated in your title. Please add this information in your result section. Please provide some information about the ambulance services available to the patients. Is it provided free of cost, remains available 24x7 and what type training was given to the escort persons.

Since this paper focuses on the transfer of the infants, specific recommendations should be made in the discussion regarding improvements, e.g. how can stabilization prior to transfer be improved, what additional training do the emergency staff require, etc

Please add a segment what type of interventions were required in neonates immediately on arrival in emergency room at your center.

Also add what is the neonatal mortality rates observed at authors center for inborn babies. Is mortality same for inborn and outborns?

Line 72: Neonatal age is the most vulnerable period in human life with neonatal mortality contributing up to 47% of overall under-five deaths. Please give reference for this statement. Is this represent global figure or data from you country?

Line 74-75: Most of these conditions, up to 75% are preventable through effective equitable measures such as early detection and timely management. Most of these conditions, up to 75% are preventable through effective equitable measures such as early detection and timely management. Please add reference.

Line 128: Since data was collected from referral cards, please give the details of the data/parameters recorded in referral cards

Line 183-184: Blood circulation was monitored 184 in 37 (39.4%) of the transferred neonates. Please elaborate how blood circulation was measured during transfer.

Best wishes,

6. PLOS authors have the option to publish the peer review history of their article (what does this mean?). If published, this will include your full peer review and any attached files.

Reviewer #1: No

Reviewer #2: No

Reviewer #3: No

Reviewer #4: No

---

## [Author Response · Author response to Decision Letter 0]

17 Mar 2022

PONE-D-21-40535

Referral Challenges and Outcomes of Neonates Received at Muhimbili National Hospital, Dar es Salaam, Tanzania.

PLOS ONE

March 17, 2022

Dear editors,

We are grateful to have received your email dated 2nd February 2022, containing the reviewers’ comments pertaining our manuscript titled “Referral Challenges and Outcomes of Neonates Received at Muhimbili National Hospital, Dar es Salaam, Tanzania” 

We are thankful to the reviewers for taking their time to assess our manuscript. We have addressed all the concerns raised by the independent peer reviewers. Those have improved the overall quality of our manuscript.

Please find below our point-by-point response, clearly indicating how and where in the manuscript changes have been made (line numbers). To assist you in readily tracking our amendments, we included a marked up copy of the manuscript that highlights changes made to the original version. We also enclose a “clean” version of the revised manuscript as supporting information.

We very much hope that this revised version of our manuscript will now be suitable for publication in PLOS medicine. We look forward to hearing from you.

Yours Sincerely,

Dr Mpokigwa Kiputa

Reviewer #1: Referral Challenges and Outcomes of Neonates Received at Muhimbili National Hospital, Dar es Salaam, Tanzania

PONE-D-21-40535

I would like to congratulate the authors for a good work done. I have some minor concerns which may improve upon the quality of the manuscript.

Thank you for the congratulatory note.

1. Introduction: lines 72-73 stated “About half of neonates die within the first 24 hours after birth and 75% in the first seven days of life” Please check this information and present it appropriately. You mean; about half of all neonatal deaths occur within the first 24 hours……

In light of your comment we have improved and updated the introduction part.

2. Participant recruitment and data collection: line 126 stated that: “Data was collected day and night…” Were there any difference in outcome when day or night referrals were considered?

The statement sought to address the fact that neonates were recruited 24/7. Based on your comment we have reanalyzed the data and found out there was no significant difference in outcome during the day Vs night referral. However, we didn’t add this factors in the analysis as we didn’t extract and record exact time of death as a variable.

3. Neonatal age in days at time of admission was not reported. This is important to understand the outcome measure presented in the study.

Categories of neonatal age in days at the time of admission has been updated in Table 1: Socio-demographic and clinical characteristics of study participants.

4. Results section monitoring during transport: lines 183-184: “Blood circulation was monitored in 37 (39.4%) of the transferred neonates…..” How was it done and how was this assessed.

Perfusion monitoring was assessed by asking the escorting health personnel if he/she checked the capillary refill time as part of monitoring during transport. The questionnaire is shared as supplementary materials and the methodology section updated accordingly.

5. Table 1: presented “Apgar score 1st and 5 minutes” however only single measurement was presented. A little explanation would dispel the confusion.

On the contrary it was two measurement presented e.g. (7-10), meaning the Apgar score was 7 in the 1st minute and 10 in the 5th minute. Table 1 has been revised to clarify the information. Thank you for noting this confusion.

Reviewer #2: General comment

This would have been an important study that contributes to the understanding of the Referral Challenges and Outcomes of Neonates Received at Muhimbili National Hospital, Dares Salaam, Tanzania. However, there are minor flaws in the method of the study, discussion and grammar that need a revision before publication.

Thank you for appreciating the importance of this study. The authors have addressed all comments and updated the manuscript accordingly. 

1. In abstract section, it needs more elaboration in why the study is needed/ gap of the study, as much as possible avoid acronyms in the abstract and rewrite the abstract in very short and precise way particularly in the result section.

Elaboration on why the study is needed has been added, acronyms have been deferred and the abstract section have been re written as suggested. The results section in the abstract has been shorten.

2. Pay great attention for punctuations.

Point noted. The revised manuscript took great consideration on punctuation. 

3. In introduction section, would start with the global and regional/ continent evidence on neonatal mortality and relate it with how referral contributes. One paragraph which describes why this study is needed. Generally, introduction lacks coherence of ideas, world to Tanzania figure of neonatal mortality, gap of the study, clear description of clinical outcome of referral neonates in previous study. Introduction needs substantial modification.

Modification have been made as suggested with a focused introduction highlighting the burden of neonatal mortality, referral care gaps, and significance of this study. Also we have included factors identified in previous studies to pose referral challenges and how they are associated with neonatal mortality. 

4. In materials and methods

Study area: Add the number of physician, nurses and other health care professionals working in neonatal ward and NICU. It gives an insight for why high burden of neonatal death; is it because of shortage of HCW or not? You stated the number of admission but we can’t get patient-to-physician proportion. 

The number of staff, cadre working at neonatal unit and NICU has been added in materials and methods section to provide an insight on the NICU capacity.

5. In line 147, it states as “All enrolled neonates were assessed at 48 hours post admission to capture the early outcome that was defined as survived or died” what is your baseline reference? If there is, cite it.

Regarding the early outcome assessment, we had no baseline reference to refer but we have included overall mortality at Muhimbili tertiary hospital. 

6. Result

In line 157, it states as “A total of 349 referred neonates were screened during the study period, one was excluded from the study due to a congenital anomaly which was incompatible with life (anencephaly).” How it could be? You already excluded the known congenital anomalies before including in the study

This is a valid point. We changed the statement to read as follows;

A total of 349 referred neonates were recruited during the study period, one was excluded from the study due to a congenital anomaly which was incompatible with life (anencephaly).

In line 161, “258 (74.1%) were delivered by Spontaneous Vertex Delivery (SVD) while 85 (24.4%) were born via caesarean section.” Rewrite this.

Thank you for pointing this, the sentence has been rewritten.

What is the rationale behind to use median/mean and IQR and SD? You used median for GA and mean for BWT….

Mean was used for data with normal distribution and median for skewed data like gestational age.

In line 176, it states as “thirty-two (9.4%) were escorted by the family member.” In method section, you stated that the pre transfer data and transportation data/monitoring during transportation were taken from accompanying personnel/escort personnel. My question is how did you collect data from those who escorted by the family member? They don’t know about the pre transfer treatments and monitoring process.

The pre transfer data were collected from the referral notes/other documents. Those escorted by family members were not in a position to provide information on monitoring during transportation, hence not applicable. Only health care personnel were asked about the pre transport and monitoring during transport.

In line 208-211, Twenty-one (67.7%) neonates with prolonged capillary refill time (CRT) died compared to ten (32.3%) who survived (OR=4.76, 95%CI (1.80-12.58), p=0.002)”. This sentence is hard to follow. Consider revision. Either describes using odds ratio or proportion in the whole document.

Revision have been done and odds ratio used to compare the outcome. Most of the text has been changed in the manuscript considering your comments.

7. The discussion is inadequate and need to be revised thoroughly. Your independent variables were not well stated as compared to other findings with possible justification. Contributing factors are not discussed in detail as compared to previous studies. It doesn’t tell us the discovery and how these results contribute to the current debate and policies in the research specific context and beyond.

Revisions have been made to the discussion section. This study has highlighted the coverage gap during the referral of a newborn and how poor clinical status on admission influence the outcome. We have explained these facts together with the challenges which have not been reported before in our setting. It’s our hope the publication of this findings will influence policies in the region and beyond.

In table 2: you used univariate or bivariate? Which one do you think is appropriate? What is difference b/n multivariate analysis and multivariable analysis? Which one is appropriate for your study?

In Table 2 we used univariate analysis taking one independent variable at a time in relation to the outcome. Multivariate analysis provides more objective approach for studying the effect of covariates on the binary outcome as fitted in the model. It addresses both categorical and continuous covariates. Model selection increases power of detection, from that point of view multivariable analysis is appropriate to be used in our study.

Generally, give much concern for grammar, punctuation, capitalization and flow of ideas.

The comment has been taken. In our revised manuscript we have given much attention to grammar, punctuation and flow of ideas.

Reviewer #3: 

The study titled Referral Challenges and Outcomes of Neonates Received at Muhimbili national hospital, Dar es Salaam, Tanzania explored problems for preparation, care during transport and outcome in terms of death within 48 hours, a crucial issue that are under addressed in countries like Tanzania. These findings will guide to plan for way forward to achieve SDG.

Thank you for the encouraging remarks.

In line 180, spelling mistake, seems it will be taxi. Others seem good

The spelling mistake corrected accordingly in the manuscript.

Reviewer #4

This manuscript describes the outcome of neonates referred to tertiary care hospital and identification of underlying factors for poor outcome. The topic is very important particularly from point of view of high NMR in developing countries. However, there are some serious flaws in the paper which need to rectify before further processing:

Thank you for acknowledging the importance of the topic, we have addressed all comments provided.

1. First of all, the authors need to work on the English language written in the paper for grammar, and syntax of the sentences.

In our revised manuscript we have given great attention to details regarding grammar and syntax of the sentences.

2. Please clarify that whether all babies were born at hospitals, if it is so, then, please elaborate the infrastructure available at these peripheral hospitals where child births have taken place. Please add whether these neonates were referred from a single health facility or multiple hospitals.

In Tanzania, most women deliver at health facility (83% as per One plan III report, 76% as per DHIS2 (2018)). In this study, we didn’t enquire on specific delivery place but in most cases babies are sent to health centers for assessment and vaccination post-delivery. 

It was noted that few babies at the time of admission were escorted by family members (9.4%). Majority of neonates were referred from health facilities; Figure 1 shows the flow of study participants referred from various districts of Dar es Salaam. The details of infrastructure/capacity of peripheral hospitals has been added in the introduction and methodology section.

3. Referral indications: Please describe the approach used to determine indication for referral. Please explain whether this information was extracted from cards with details of parameters used in referral cards. Sometime, the indications listed are not necessarily mutually exclusive” e.g. need for mechanical ventilation may overlap with prematurity or birth asphyxia. If there was more than one indication, how was this treated analytically?

The information regarding indication for referral was extracted from referral notes. This information was captured at the time of admission before any further history was taken. We noted poor documentation of pre referral information including pre referral care provided. 

Figure 2 shows the coverage gaps on what parameters have been documented as measured. 

Nationally, we still have a gap on harmonization of documentation used for neonatal care. The ongoing Newborn Essential Solutions and Technologies (NEST) program is working towards achieving this. This might be considered as one of our weaknesses but Figure 2 shows pre transport care and the clinical admission status of all neonates received using the TOPS model rather than admission diagnoses. These are common preventable conditions which contributes to neonatal mortality and has to be checked and tackled in every admission/referral.

4. What definitions did you use for classification of period of gestation, birth asphyxia, and severity of hypothermia (like moderate or severe?)

Gestational age is frequently poorly recorded and/or women present late to care. The accurate determination of gestational age or classification of prematurity is critically important as your results present approximately 40% prematurity. Please identify this specifically for weight as well – it is extremely important to know if these were scale weights at birth and what was done for home births (if any).

The cut points used for gestation age and hypothermia were according to the WHO definition. The assessment of Birth Asphyxia was out of the scope of our study, we didn’t utilize clinical diagnoses in our analyses.

Gestational age was calculated by assessing the date of the last normal menstrual period or using the obstetric ultrasound report if available. Then this information was compared to the one recorded in the Reproductive and child health clinic card. 

The birth weight reported were all scale weight from the respective facilities. The validation of birth weight measurement is not within the scope of this study. The results of previous study conducted in Tanzania ‘EN BIRTH study’; can provide further information on coverage and quality measurements in routine information system.

5. The authors have not included any information about the comparison between the referred neonates that lived and those that died. Please add information that describes this comparison.

Thank you for the comment, we compared the referred neonates that lived Vs those died in terms of clinic status at admission. Figure 3 shows the combined effect of clinical status at admission and neonatal outcome.

6. Do you have any information about timeliness of referrals or care provided before referrals? In final discussion and conclusions, you have mentioned that the peripheral center and pre-referral stabilization is needed as well as upgrades. While this may be true, you do not present sufficient information in the manuscript to lead to this recommendation. Also explain about the duration of stay for neonates at referring facility and treatment provided before referral. Please add duration transport as well distance travelled by neonates and perform statistical analysis of this factor also.

Pre referral care has been presented in the result section and summarized using the TOPS model in Figure 2. The timeliness of referral, the duration of stay at referring facilities, duration of transport and the distance travelled were difficult to ascertain in our study as they were not recorded anywhere. The information we collected were within the provision of care provided. To analyze the above factors, we will need to design another study which cut across the whole referral network system including assessment at the peripheral facilities. 

The comment has been taken for designing further studies.

7. As the sample size 348, in order to know if this sample is representative for the whole study population, it would be needed more information regarding hospital and setting. What is the population size attended by this hospital, how many neonatal admissions by year this hospital has and how many are inborn and how many outborn. This would give us a roughly idea if the sample size is enough to be representative for the whole population. Also what is the NMR in this hospital or at least in the region (authors have given the national NMR but likely the regional is different). Please also write exact number of the neonates at every place where percentage is used.

The annual neonatal admission by year ranges between 1440 and 3360 neonates. This has been reported under the material and methods section. There is currently no study which has established the NMR at Muhimbili National Hospital or Dar es Salaam region. The crude data from the ministry of health reports, 2018- 2019 shows Dar es Salaam region being the number one contributor of neonatal mortality with an average neonatal mortality of 14.6 per 1000 facility births. 

In a revised manuscript the exact number of neonates have been written in every place where percentage is used.

8. Explain in a better way how the factors affecting the outcome of neonates were identified. What kind of analysis was done to determine significance of parameters? The significance level indicated was for what? Whether only those parameters having significant values on univariate analysis were included in multivariate analysis?

From a univariate analysis, factors with P value < 0.2 were included in the final multivariable model. The significance level was indicated for outcome defined as death, hence only those parameters with significant value on univariate analysis were included in multivariable analysis.

9. In LMICs, antenatal screening is one of the important factors in identification of high risk pregnancies, but surprisingly this information was missed in manuscript.

We agree about the importance of antenatal screening but the information is beyond the scope of our study. It could be true that high risk pregnancies have been referred early but with high number of facility deliveries, strengthening referral system is needed.

10. There are no details of referral challenges that might be faced during referral of neonates in the region as you have stated in your title. Please add this information in your result section. Please provide some information about the ambulance services available to the patients. Is it provided free of cost, remains available 24x7 and what type training was given to the escort persons.

In most Government health facilities, the ambulance service is free of cost and yes it is available 24/7. This information has been added in the introduction and discussion section as needed.

Currently, there is no harmonized referral network and trainings but the escort nurses come from the respective units of referring facilities and are supposed to be trained on skills and competences at least for essential newborn care if not comprehensive care.

The referral challenges are reported in Figure 2 in the results section. The information regarding the type of training given to the escort personnel is reported in the discussion section.

11. Since this paper focuses on the transfer of the infants, specific recommendations should be made in the discussion regarding improvements, e.g. how can stabilization prior to transfer be improved, what additional training do the emergency staff require, etc

Specific recommendations regarding improvements on pre transfer care and stabilization is found in the discussion section. The whole discussion has been revised accordingly.

12. Please add a segment what type of interventions were required in neonates immediately on arrival in emergency room at your center

The segment has been added in the material and methods section. It reads as follows;

Neonates who were found to be hypoglycemic on admission were given bolus intravenous dextrose, those who were hypoxic received supplemental oxygen according to their needs and those who were hypothermic were kept in a warmer. Neonates who were in shock upon arrival were resuscitated with intravenous fluids according to the hospital protocol. There was room for escalation of care for neonates who needed advanced respiratory support such as ventilating machine.

Additionally, the clinical admission status found in the cases have been presented in the results and discussed in the manuscript. 

13. Also add what is the neonatal mortality rates observed at authors center for inborn babies. Is mortality same for inborn and outborns?

The information has been added in the introduction section, it reads as follows;

According to raw data from the MNH, neonatal mortality among the out-borns is almost twice as higher compared to the in-borns admitted at the same neonatal unit.

14. Line 72: Neonatal age is the most vulnerable period in human life with neonatal mortality contributing up to 47% of overall under-five deaths. Please give reference for this statement. Is this represent global figure or data from you country?

Reference have been provided accordingly and introduction section revised.

15. Line 74-75: Most of these conditions, up to 75% are preventable through effective equitable measures such as early detection and timely management. Please add reference.

The statement has been modified it is no longer available in the manuscript. The point has been noted.

16. Line 128: Since data was collected from referral cards, please give the details of the data/parameters recorded in referral cards

Line 135-137: Data on demographic characteristics of the study participants and pre referral treatment if any were extracted from referral cards. There are variations of the referral forms used by the referring facilities. Currently, the Newborn Essential Solutions and Technologies (NEST) program is working towards achieving this.

17. Line 183-184: Blood circulation was monitored 184 in 37 (39.4%) of the transferred neonates. Please elaborate how blood circulation was measured during transfer.

Blood circulation was monitored by checking the capillary refill time periodically. To remove the confusion, the sentence has been modified to read as follows;

Capillary refill time was monitored in 37 (39.4%) of the transferred neonates while Random blood glucose was checked in 27 (28.7%).

---

## [Decision Letter · Decision Letter 1]

18 Apr 2022

PONE-D-21-40535R1Referral Challenges and Outcomes of Neonates Received at Muhimbili National Hospital, Dar es Salaam, Tanzania.PLOS ONE

Dear Dr. KIPUTA,

Thank you for submitting your manuscript to PLOS ONE. After careful consideration, we feel that it has merit but does not fully meet PLOS ONE’s publication criteria as it currently stands. Therefore, we invite you to submit a revised version of the manuscript that addresses the points raised during the review process.**Please, ensure that you integrate responses to reviewers' comments (previous and current comments) and current into the manuscript itself (not only in the response letter).   **Please, acknowledge study other study limitations that were highlighted during the review process.Please, make sure that the study conclusion is based on study findings. Please submit your revised manuscript by Jun 02 2022 11:59PM. If you will need more time than this to complete your revisions, please reply to this message or contact the journal office at plosone@plos.org. Please include the following items when submitting your revised manuscript:A rebuttal letter that responds to each point raised by the academic editor and reviewer(s). You should upload this letter as a separate file labeled 'Response to Reviewers'.A marked-up copy of your manuscript that highlights changes made to the original version. You should upload this as a separate file labeled 'Revised Manuscript with Track Changes'.An unmarked version of your revised paper without tracked changes. You should upload this as a separate file labeled 'Manuscript'.If applicable, we recommend that you deposit your laboratory protocols in protocols.io to enhance the reproducibility of your results. Protocols.io assigns your protocol its own identifier (DOI) so that it can be cited independently in the future. For instructions see: https://journals.plos.org/plosone/s/submission-guidelines#loc-laboratory-protocols. Additionally, PLOS ONE offers an option for publishing peer-reviewed Lab Protocol articles, which describe protocols hosted on protocols.io. Read more information on sharing protocols at https://plos.org/protocols?utm_medium=editorial-email&utm_source=authorletters&utm_campaign=protocols.

We look forward to receiving your revised manuscript.

Kind regards,

Elsayed Abdelkreem, MD, PhD

Academic Editor

PLOS ONE

Journal Requirements:

Reviewers' comments:

Reviewer's Responses to Questions

**Comments to the Author**

1. If the authors have adequately addressed your comments raised in a previous round of review and you feel that this manuscript is now acceptable for publication, you may indicate that here to bypass the “Comments to the Author” section, enter your conflict of interest statement in the “Confidential to Editor” section, and submit your "Accept" recommendation.

Reviewer #1: All comments have been addressed

Reviewer #4: All comments have been addressed

2. Is the manuscript technically sound, and do the data support the conclusions?

Reviewer #1: Yes

Reviewer #4: Yes

3. Has the statistical analysis been performed appropriately and rigorously? 

Reviewer #1: Yes

Reviewer #4: Yes

4. Have the authors made all data underlying the findings in their manuscript fully available?

Reviewer #1: Yes

Reviewer #4: Yes

5. Is the manuscript presented in an intelligible fashion and written in standard English?

Reviewer #1: Yes

Reviewer #4: Yes

6. Review Comments to the Author

Reviewer #1: All my comments have been addressed. I would like to congratulate the authors for a good work done.

Reviewer #4: Dear authors,

Thank you for addressing my queries in detail and submission of revisions. Authors have improved the manuscript very nicely, but I have some minor issue:

1. Although grammar part had been improved but please look on use of unnecessary capital words. Line 166: Model, line 185: Spontaneous Vaginal Delivery.

2. Main issue is with the inadequacy of the data analysis for primary aim of the study.

In abstract part (background section, line 24): Authors claim “This study assessed the pre-referral care……”.

However, as per the information provided in result section (line 189), no details of prereferral care have been given by the authors. The information just describes the numbers of neonates having IV-line insertion, temperature measurement, oxygen saturation and blood glucose monitoring.

Same point had been highlighted in the response letter points number 3 and 6 regarding inadequacy of documentations in referral cards.

I will suggest author to add this point (i.e. inadequate assessment of prereferral care) in limitations sections, please.

3. In abstract part, result section line 44-49 need to be rephrased while reporting odd’s ratios and p-value. Please avoid repeated use of odds of dying or likely to die repeatedly as this can be easily understandable from odd’s ratio against each variable.

4. In conclusion part, line 58-59; Please shift this line from conclusion to discussion part. This is not a finding of this study but may serve as future recommendation.

5. Line 213: I request to the authors to state indications of referral more explicitly rather than mentioning “seek specialized care, lack of newborn unit or lack of equipment”, which are very vague.

6. As authors have acknowledged in response letter at points number 3, 6 and 9th, there are some limitations in recording full data. So, please add these statements in the limitations section at the end of manuscript.

7. Please avoid repetition of the facts “no ambulance had an incubator” at line 211-212, line 222.

Best wishes,

7. PLOS authors have the option to publish the peer review history of their article (what does this mean?). If published, this will include your full peer review and any attached files.

Reviewer #1: No

Reviewer #4: No

---

## [Author Response · Author response to Decision Letter 1]

17 May 2022

PONE-D-21-40535R1

Referral Challenges and Outcomes of Neonates Received at Muhimbili National Hospital, Dar es Salaam, Tanzania.

PLOS ONE

May 17, 2022

Dear editors,

We are grateful to have received your email dated 19th April 2022, containing the reviewers’ comments pertaining our manuscript titled “Referral Challenges and Outcomes of Neonates Received at Muhimbili National Hospital, Dar es Salaam, Tanzania” 

We are appreciative to the reviewers for taking their time to assess our manuscript a second time after our first review response. We have addressed all the concerns raised by the independent peer reviewers and improved the overall quality of our manuscript.

Please find below point-by-point response, clearly indicating how and where in the manuscript changes have been made (line numbers). To assist you in readily tracking our amendments, we included a marked up copy of the manuscript that highlights changes made to the original version. We also enclose a “clean” version of the revised manuscript. All comments from the reviewers (previous and current comments) have been integrated into the manuscript itself (not only in the response letter). Study limitations, conclusion and recommendations revised accordingly. Reviewers’ responses are highlighted in the manuscript. 

Additional edits to reformat the manuscript have been done and can be observed with track changes and summarized in the response to comments letter specifying sections and line numbers as per no. 8 below. Kindly note that, these edits do not alter any information as per reviewers comments but rather improve the organization and grammar of the manuscript.

We very much hope that this revised version of the manuscript will now be suitable for publication in PLOS medicine. We look forward to hearing from you.

Yours Sincerely,

Dr Mpokigwa Kiputa

Journal Requirements: 

Thank you for the comment, reference list has not been updated and remains as per previous edits according to initial reviewer’s comments.

Reviewer #1: All my comments have been addressed. I would like to congratulate the authors for a good work done.

Thank you for the congratulatory note. We really do appreciate your time invested to improve the quality of the manuscript.

Reviewer #4: Dear authors, Thank you for addressing my queries in detail and submission of revisions. Authors have improved the manuscript very nicely, but I have some minor issue:

Thank you for the positive remarks

1. Although grammar part had been improved but please look on use of unnecessary capital words. Line 166: Model, line 185: Spontaneous Vaginal Delivery.

Thanks for noticing. The capitalization errors have been addressed accordingly in line 169 and line 188 in the manuscript. The whole manuscript was revised and other font edits have been made in line 146, 147 and 158.

2. Main issue is with the inadequacy of the data analysis for primary aim of the study.

In abstract part (background section, line 24): Authors claim “This study assessed the pre-referral care……”

However, as per the information provided in result section (line 189), no details of pre referral care have been given by the authors. The information just describes the numbers of neonates having IV-line insertion, temperature measurement, oxygen saturation and blood glucose monitoring.

Same point had been highlighted in the response letter points number 3 and 6 regarding inadequacy of documentations in referral cards.

I will suggest author to add this point (i.e. inadequate assessment of pre referral care) in limitations sections, please.

Pre referral care entails stabilization of the sick neonate prior to transfer according to the S.T.A.B.L.E (Sugar, Temperature, Airway, Blood pressure, Lab work and Emotional support) program. It was developed by Kristine Karlesen in 1980s as part of research project. Henceforth adopted worldwide as a standard practice. 

Establishing an IV access, temperature measurement, oxygen saturation and blood glucose monitoring is part of the pre referral care as it was laid out in our study using the TOPS model. Although we agree that to a certain extent it was inadequate given the fact that not all parameters were assessed and no direct assessment was carried out at the referring centers. In this study, pre referral care assessment was based on documentation provided from the referral notes.

In limitations section line 347-349, this point has been added on and it now reads as follows;

Limitation in recording full data due to poor/incomplete documentation based on the referral forms, consequently contributed to the inadequate data of pre-referral care assessment, reasons for referral and monitoring enroute information

In addition, the methodology section in the abstract has been updated accordingly, please refer to line 29, 30 and 34 to match line 137 -139 in the main manuscript (highlighted yellow for easier tracing)

3. In abstract part, result section line 44-49 need to be rephrased while reporting odd’s ratios and p-value. Please avoid repeated use of odds of dying or likely to die repeatedly as this can be easily understandable from odd’s ratio against each variable

The statement in result section line 45-51 has been rephrased and it now reads as follows;

Hypothermic neonates had an increased chance of dying compared to those who were normothermic (OR=2.09, 95% CI (1.05-4.20), p=0.037). The chance of dying among those presenting with hypoxia was almost three times (OR=2.88, 95%CI (1.44-5.74), p=0.003) while those with poor perfusion was almost five times (OR=4.76, 95%CI (1.80-12.58), p=0.002). Additionally, neonates who had hyperglycemia (RBG > 8.3mmol/l) on arrival had a higher probability of dying compared to those who were euglycemic [(OR= 3.10, 95% CI (1.19 – 8.09) p=0.021].

4. In conclusion part, line 58-59; please shift this line from conclusion to discussion part. This is not a finding of this study but may serve as future recommendation.

We agree with this observation. The line aforementioned has been removed from the conclusion part and added to recommendation line 355-356 in the main manuscript

5. Line 213: I request to the authors to state indications of referral more explicitly rather than mentioning “seek specialized care, lack of newborn unit or lack of equipment”, which are very vague.

The indications for referral were extracted from the referral notes at the time of admission before any further history was taken. Unfortunately, the reasoning were clustered and no much information provided. This has been addressed as one of the limitations of our study. Documentation using standard harmonized forms is a critical and currently NEST 360 program is working on improving all forms including introducing inpatient neonatal register.

The study limitation has been updated to accommodate this information and it reads as 

Limitation in recording full data due to poor/incomplete documentation based on the referral forms, consequently contributed to the inadequate data of pre-referral care assessment, reasons for referral and monitoring enroute information. Please refer to line 347 – 349.

6. As authors have acknowledged in response letter at points number 3, 6 and 9th, there are some limitations in recording full data. So, please add these statements in the limitations section at the end of manuscript.

The statement has been added accordingly in line 347-349 in the limitation section. It reads as follows;

Limitation in recording full data due to poor/incomplete documentation based on the referral forms, consequently contributed to the inadequate data of pre-referral care assessment, reasons for referral and monitoring enroute information. Please refer to line 344 – 346 in the manuscript.

7. Please avoid repetition of the facts “no ambulance had an incubator” at line 211-212, line 222.

The aforementioned line 211 which now stands as line 214 had been modified accordingly to avoid repetitions. The edits have been made in the manuscript.

8. Additional edits of grammar, font and some wordings have been made in the manuscript for the purpose of clarity. They are on track changes and summarized per section and line numbers below. All the edits are cosmetic and do not alter previously revised manuscript.

Abstract – line 26, 34, 41, and 60

Introduction – line 105

Materials and Methods - line 120, 123, 124, 146, 147, 158, 169, 175 and 177

Results – 185, 192, 193 and 228

Discussion – 251, 275, 285, 290, 302, 305, 325, 326, 339, 352 and 354

---

## [Decision Letter · Decision Letter 2]

23 May 2022

Referral Challenges and Outcomes of Neonates Received at Muhimbili National Hospital, Dar es Salaam, Tanzania.

PONE-D-21-40535R2

Dear Dr. KIPUTA,

We’re pleased to inform you that your manuscript has been judged scientifically suitable for publication and will be formally accepted for publication once it meets all outstanding technical requirements.

Kind regards,

Elsayed Abdelkreem, MD, PhD

Academic Editor

PLOS ONE

Additional Editor Comments (optional):

Reviewers' comments:

Reviewer's Responses to Questions

**Comments to the Author**

1. If the authors have adequately addressed your comments raised in a previous round of review and you feel that this manuscript is now acceptable for publication, you may indicate that here to bypass the “Comments to the Author” section, enter your conflict of interest statement in the “Confidential to Editor” section, and submit your "Accept" recommendation.

Reviewer #4: All comments have been addressed

2. Is the manuscript technically sound, and do the data support the conclusions?

Reviewer #4: Yes

3. Has the statistical analysis been performed appropriately and rigorously? 

Reviewer #4: Yes

4. Have the authors made all data underlying the findings in their manuscript fully available?

Reviewer #4: Yes

5. Is the manuscript presented in an intelligible fashion and written in standard English?

Reviewer #4: Yes

6. Review Comments to the Author

Reviewer #4: Dear authors,

Thank you for revising the manuscript and addressing my queries. I am satisfied with the revisions submitted. Improving the neonatal survival in a low resource settings is major area of concern for the health policy makers all over the world. Authors need to be commended for addressing this topic.

Thank you,

Best wishes,

7. PLOS authors have the option to publish the peer review history of their article (what does this mean?). If published, this will include your full peer review and any attached files.

Reviewer #4: No

---

## [Editor Report · Acceptance letter]

27 May 2022

PONE-D-21-40535R2 

Referral Challenges and Outcomes of Neonates Received at Muhimbili National Hospital, Dar es Salaam, Tanzania 

Dear Dr. Kiputa:

I'm pleased to inform you that your manuscript has been deemed suitable for publication in PLOS ONE. Congratulations! Your manuscript is now with our production department. 

Kind regards, 

on behalf of

Dr. Elsayed Abdelkreem 

Academic Editor

PLOS ONE